# Multilayer Reflective Coatings for BEUV Lithography: A Review

**DOI:** 10.3390/nano11112782

**Published:** 2021-10-20

**Authors:** Paul C. Uzoma, Salman Shabbir, Huan Hu, Paul C. Okonkwo, Oleksiy V. Penkov

**Affiliations:** 1ZJU-UIUC Institute, International Campus, Zhejiang University, Haining 314400, China; upaul@intl.zju.edu.cn (P.C.U.); salmanshabbir862@gmail.com (S.S.); huanhu@intl.zju.edu.cn (H.H.); 2Department of Polymer and Textile Engineering, Federal University of Technology, P.M.B. 1526, Owerri 460114, Nigeria; 3Mechanical & Mechatronics Engineering Department, College of Engineering, Dhofar University, Salalah 211, Oman; pokonkwo@du.edu.om; 4Department of Mechanical Science and Engineering, The University of Illinois at Urbana-Champaign, Urbana, IL 61801, USA

**Keywords:** BEUV lithography, multilayer mirrors, X-ray optics, reflectivity

## Abstract

The development of microelectronics is always driven by reducing transistor size and increasing integration, from the initial micron-scale to the current few nanometers. The photolithography technique for manufacturing the transistor needs to reduce the wavelength of the optical wave, from ultraviolet to the extreme ultraviolet radiation. One approach toward decreasing the working wavelength is using lithography based on beyond extreme ultraviolet radiation (BEUV) with a wavelength around 7 nm. The BEUV lithography relies on advanced reflective optics such as periodic multilayer film X-ray mirrors (PMMs). PMMs are artificial Bragg crystals having alternate layers of “light” and “heavy” materials. The periodicity of such a structure is relatively half of the working wavelength. Because a BEUV lithographical system contains at least 10 mirrors, the optics’ reflectivity becomes a crucial point. The increasing of a single mirror’s reflectivity by 10% will increase the system’s overall throughput six-fold. In this work, the properties and development status of PMMs, particularly for BEUV lithography, were reviewed to gain a better understanding of their advantages and limitations. Emphasis was given to materials, design concepts, structure, deposition method, and optical characteristics of these coatings.

## 1. Introduction

X-ray optics is one of the key technologies in various scientific, engineering, and industrial applications. This technology has attracted intensive attention worldwide due to its importance [1,2,3,4]. Developing a local technological chain to design and manufacture high-reflective X-ray optical components is crucial for national economics and security. These components are vital for various scientific and technical fields such as lithography, high-resolution microscopy, X-ray fluorescence analysis, synchrotrons, free-electron lasers, and space astronomy.

Among these, one of the most critical applications is next-generation lithography. In the microelectronics industry, there is an unceasing global trend of scaling down the manufacturing procedures to increase the operating frequency and decrease the power consumption, and of enhancing the computational capacity of microprocessors (Figure 1) [5,6,7]. Thus, increasing the resolution of the lithography equipment employed in the industry and the associated decrease in the working wavelength of the required light beam is the main technical route agreed upon by industry and academia [8].

Lately, the resolution has been enhanced by substituting the deep ultraviolet (DUV) wafer scanners with novel devices that use soft X-ray radiation, also called extreme ultraviolet (EUV) radiation at λ = 13.5 nm [5]. A further reduction in the working wavelength to 6.7 nm and the use of beyond extreme ultraviolet (BEUV) radiation will boost the performance of microprocessors even more [5,9]. Significant efforts were directed into this direction by ASML, which is the world’s largest manufacturer of DUV and EUV lithographical equipment [10,11,12,13,14].

Soft X-rays exhibit a higher degree of absorption in all materials, which has necessitated the use of reflective optics. Thus, the reflectivity of X-ray optics is crucial for the efficiency of lithographical stepper machines. Figure 2 illustrates the optical system of the typical stepper machine used in lithography [15]. As can be seen, it contains at least three mirrors in the illuminator optics and six mirrors in the projection optics sections. The overall reflectivity of such an optical system is approximately one hundred-fold lower than the reflectivity of a single mirror due to the multiple reflections [16,17]. Increasing each mirror’s reflectivity from 53 to 65% will increase the integral throughput of the system approximately seven-fold. Thus, even a minimal improvement in reflectivity gives an immediate boost to the performance of lithographical systems, which includes the spatial resolution, EUV light intensity, and speed. In addition, it reduces the overall production cost by decreasing the exposition time.

The weak reflection of soft X-ray radiation by solids demands advanced reflective optical equipment. Immediately after the discovery of X-rays, it was stated that the refractive index of all materials is very small (∼10^−5^) [18]. Later, by discovering X-ray diffraction from crystals, the method of deflecting X-rays was provided for the first time [18]. A new version of X-ray diffraction has evolved over the past 25 years by building well-defined artificial diffracting structures instead of natural crystals to manipulate X-rays. Usually, such artificial structures consist of periodically alternated nanolayers of two materials, as shown in Figure 3. Thus, the reflectivity of X-rays is amplified due to the multiplication of interfaces. Such structures are called periodical multilayer X-ray mirrors (PMMs).

PMMs are known as artificial Bragg crystals having alternate layers of “light” and “heavy” materials (Figure 2); they are also called “spacer” and “absorber.” The periodicity of such a structure is almost half of the working wavelength when the angle of incidence (AOI) is near 90° [17,19]. PMMs realize high reflectivity by the constructive interference of light reflected from the interfaces of assembled layers of different optical contrasts. It is important to note that the refractive index in the X-ray range is complex and equals 1 − *δ* + iβ, where *δ* is the refractive index decrement and β represents the absorption index. Therefore, at any given wavelength, materials with a high *δ* and a low β act as a reflective layer, whereas those with a low *δ* and a low β serve as spacer layers. An array of layers with high and low *δ* (alternated reflectors and spacers in a stack) provides optical contrast, as shown in Figure 3. Both the reflector and spacer layers need a low β to minimize any loss of light by absorption. For instance, at a wavelength of above 12.8 nm, molybdenum and silicon can serve as a pair of reflector/spacer materials and work together to serve as PMMs.

## 2. Materials and Design

The design of high-reflective PMMs for the specific wavelength includes two essential steps. The first step is the choice of a material pair. The selection is based on the optical properties of materials in the given wavelengths range. For example, Mo/Si PMMs are primarily used for the wavelength of 13.5 nm (EUV), and Mo/B and La/B PMMs are excellent for 6.7 nm (BEUV) because boron has lower absorption in this range compared to silicon [20,21,22,23,24]. Co/C PMMs are the most effective for the shorter wavelength range of 4.4–4.5 nm [25].

The second step is selecting the thickness of the individual layers and the number of layers. This step is generally based on computer simulations [18]. For instance, for BEUV radiation and 5° angle of incidence off normal, the thicknesses of B (light) and Mo (heavy) layers should be around 2.25 and 1.12 nm, respectively. Moreover, for reflectivity at a near-normal angle of incidence (AOI), La/B4C PMMs taken as very high reflective coatings operating at λ = 6.65 nm mostly require 200 periods of B_4_C layers with 2.0 nm thickness B_4_C and La layers with 1.6 nm thickness.

For the first time, the material-based design of PMMs for the 2–12 nm wavelength range was undertaken by Claude et al. [22]. It was revealed that Mo/Si PMMs are not suitable for the BEUV range due to the high absorption of Si mentioned above. Instead, following their calculations, Mo/B PMMs should show high reflectance at 6.6–11.5 nm wavelengths. A plethora of studies has been carried out on Mo/B_4_C, Pd/B_4_C, and La/B_4_C PMMs that use boron carbide (B_4_C) as a replacement for B [26,27,28,29,30].

In various studies, boron has been identified as the best material for the spacer layer for the BEUV range. However, B-based PMMs have been seldomly investigated despite their superior theoretical performance compared to other materials. The discrepancy in the volume of study is ascribed to the complexity of the deposition of the high-quality B-based PMMs. Similar to the manner in which boron carbide films are nicely produced by magnetron sputtering, pure boron is challenging to sputter due to its low atomic number [31]. The complication in the production of B-based PMMs is also caused by the dielectric nature of boron. An extended-time magnetron sputtering of the dielectric target can cause instability of the deposition process because of the electrical charging of surfaces. In the case of B_4_C, the behavior of the deposition procedure was more certain because it is easier to deposit.

As a result, for Mo/B PMMs produced by e-beam evaporation, the experimentally determined reflectivity was less than 10% [22]. Thus, Boron in the PMMs was replaced mainly by the less efficient B_4_C. For comparison, the best B_4_C-based PMMs, such as La/B_4_C, Mo/B_4_C, and Sb/B_4_C, demonstrated the reflectivity of around 20–40% at the wavelength of 6.6 nm [26,32,33]. In the recent study published by Zhu et al. in 2020, the peak reflectivity of 10% was recorded for Mo/B_4_C PMMs [34]. In general, the low reflectivity of B_4_C-based PMMs should be attributed to the relatively high absorption of carbon at the 6.6 nm wavelength and various structural imperfections.

## 3. Structural and Optical Properties

The nanostructure of the PMMs has a pivotal impact on optical behavior. The nanostructure can be controlled by a comprehensive understanding of the relationship between the structure and deposition conditions. The structural defects, such as interface roughness and intermixing between layers, will decrease the interfaces’ sharpness and reduce the overall reflectivity [35,36]. Thus, without the knowledge of the real structure, it is impossible to correctly predict the reflectivity of PMMs. Hence, multiple studies in this field have been performed with a particular focus on the structure of PMMs such as Sc/Si [37,38] and Co/C [39].

The intermixing occurs in many nanolayered systems because of solid-state chemical reactions. For example, the deposition of Mo on the top of Si led to the formation of a thin layer of molybdenum disilicide (MoSi_2_). In the case of Mo and B, the appearance of molybdenum diboride MoB_2_ occurred (Figure 4). Intermixing in PMMs is considered a structural imperfection because intermixed layers reduced the sharpness of the interface and deteriorated the reflectivity despite their low thickness (around 1 nm) [37]. For instance, Sertsu et al. [40] observed a peak reflectivity value of 6.65% at λ = 6.9 nm and 10° AOI for a B_4_C/CeO_2_ multilayer. However, the value was 4.4-fold lower than the calculated value due to significant interface diffusion. Therefore, suppression or even full illumination of the intermixing is crucial for the improvement of the reflectivity of PMMs. In addition, the knowledge about intermixing mechanisms in Sc/Si and Co/C PMMs developed in early studies [37,38,39] allowed suppression of the intermixing by introducing diffusion barriers and improved the temperature stability of PMMs [41,42,43].

In addition to the intermixing, the interface roughness is another factor crucial for reflectivity. It was shown that adjusting the pressure in the deposition step can significantly reduce the interface roughness in Mo/Si PMMs [45]. In addition, the roughness is also affected by the thickness of individual layers [46]. Thus, these factors should be considered simultaneously. It was demonstrated for EUV PMMs such as Mo/Si and Cr/Sc that ion polishing reduced the roughness [47,48]. Most recently, ion polishing was used for W/Si PMMs [49]. In the latter case, 0.1 to 0.5 nm of each Si layer was etched with 100 eV Ar ions. The analysis revealed that ion polishing reduces the vertical propagation of roughness from layer to layer by a factor of two and favorably affects the lateral correlation length.

Although significant efforts were directed to the investigation and improvement of B_4_C based PMMs, the structure of the Mo/B PMMs remains virtually unknown due to the earlier mentioned complications in the synthesis. The initial study on the optical properties of Mo/B PMMs was conducted in 1996, and did not include a comprehensive structural analysis [22]. Due to the difficulties of boron sputtering mentioned above, other research in this field was not performed for several decades. Recently, however, several groups reported the successful synthesis of B-based PMMs.

The manufacturing of La/B PMMs was reported by Makhotkin and co-workers [28]. Despite high expectations, these PMMs showed a low reflectivity because of the intermixing of the layers and the high interface roughness. In addition, the active chemical properties of La make it challenging to preserve the reflectivity of La-based PMMs [34]. To improve the reflectivity of La/B PMMs, the nitridation of the interfaces between La and B was recommended [10]. The nitridation of La represses the intermixing of La and B, thereby enhancing the optical contrast between La and B. Xu et al. used the nitridation process to improve Pd/Y PMMs at a wavelength of 9.2–9.3 nm [50]. In both cases, the reflectivity of the PMMs was significantly improved by nitridation. Nonetheless, periodicity errors developed because of the technical complexity of selective nitridation. These errors prevent reaching theoretical reflectivity. Thus, the deposition technique should be improved to avoid the adverse effects associated with nitridation.

### 3.1. Structure and Optical Characteristics of Molybdenum-Based BEUV PMMs

In 1996, Claude and co-workers synthesized 100 period Mo/B multilayers using a deposition system that contained a cryopump vacuum chamber with two magnetron sputtering guns of 1.5 cm diameter [22]. This mirror exhibited a peak reflectance of ~9.4% at λ = 6.67 nm and a 5° angle of incidence. According to the low-angle X-ray diffraction (LAXD) spectra, they assumed that the general structure of the Mo/B multilayer was satisfactory with an interface width of ~0.35/0.65.

Recently, Penkov et al. successfully synthesized Mo/B PMMs with the estimated reflectivity value of 53% [44]. It was shown that Mo/B PMMs deposited by magnetron sputtering are comprised of smooth, amorphous Mo and B layers divided by ~0.4 nm thick interlayers, which are a mixture of MoB and MoB_2_. The average density of the interlayers is close to that of MoB_2_. From these experimental observations, the optical performance in the BEUV range was calculated considering the real nanostructure of Mo-B interfaces. The computer modeling signified that the Mo/B PMMs should display a reflectivity of ~53% at a working λ = 6.7 nm (Figure 5) [44]. Such reflectivity is many times higher than that of one of the B_4_C structures reported by other researchers [34,40,51]. Table 1 gives a summary of the recent achievements of PMMs’ reflectivity in the BEUV range.

Regarding the Mo/B PMMs, even the expected reflectivity of 50–53% was still smaller than the theoretical maximum, which was around 70%. This difference in reflectivity was attributed to the formation of interlayers, high interface roughness, and bad structural uniformity. The first two factors reduced the sharpness of interfaces between Mo and B within the periodical stack. These factors led to the reduction of reflectivity.

The structural uniformity implies the regularity of the thickness of the layers and interface structure across the whole multilayer stack. Apparent errors in the thickness of individual layers or their continuous variation (drifting) caused by various factors deteriorate the uniformity. Preserving structural uniformity is essential for maintaining Bragg’s diffraction condition. The nature of the periodicity is defined mainly by the deposition rate’s time stability and maintenance of the substrate temperature [24]. The degree of the uniformity can be quickly evaluated by low-angle X-ray reflectometry [44], and manifests itself in the form of broadening of diffraction peaks, as shown in Figure 6. The real structure suffers from the gradual increase in the interlayers due to the increase in the substrate temperature during the deposition. The theoretical model implies the ideally periodical structure. The inset shows the broadening of the diffraction peak due to low structural uniformity.

It was reported that the structure and reflectivity of the Mo/B PMMs were susceptible to deposition temperature. Increasing the temperature led to the larger thickness of interlayers [44]. In the initial study, the temperature was not controlled; it was just recorded as the magnetron power function. Because the deposition temperature was varied during the deposition, a corresponding increase in the roughness and thickness of interlayers along the multilayer stack was expected, leading to a deterioration in the structural uniformity. As it was considered, the total reflectivity of Mo/B PMMs can be lower than the theoretical estimation. In addition, it should be noted that high structural uniformity allows effective use of computer simulation of low-angle XRD for studying interfaces with sub-nanometer resolution [18]. Applying this method gives new insight into the fundamental mechanism of low-scale interlayer mixing during the manufacturing of PMMs.

### 3.2. Structure and Optical Characteristics of Lanthanum-Based BEUV PMMs

La-based PMMs have been shown to possess a greater reflectance than Mo-based PMMs due to their bulk optical properties and are therefore preferred candidates for applications such as EUV lithography, which requires high photon transmission. La/B_4_C PMMs have also been considered for use in X-ray fluorescence spectroscopy [7], particularly for boron detection [8]. The theoretical reflectivity of La/B-based PMMS is >80% at 6.6 nm but the highest experimentally obtained reflectivity is still significantly lower. Makhotkin et al. prepared La/B PMMs with different periods (7.8–3.2 nm wavelength range) [28]. In the optical characterization of La/B PMMs, their choice of angle of incidence in respect to the surface normal was made to exhibit the maximum reflectance at 6.8 nm. The measured reflectivity was compared to the calculated maximum reflectivity that is computed for a standard 40 period La/B multilayer without interface roughness, employing the bulk values for La and B densities and the ratio of as-deposited La and B layer thickness. Their results proved that increasing the thickness of the multilayer periods can decrease the difference between the calculated and measured reflectivity, as shown in Figure 7. In addition, they showed that the calculated reflectance values can be achieved experimentally by improving the multilayer interface quality.

Platonov and co-workers compared the structural and optical performance of La/B_4_C and La_2_O_3_/B_4_C multilayers [52]. The measured maximum peak reflectivity of La/B_4_C is 48.9% at λ = 6.68 nm, and La_2_O_3_/B_4_C showed 39.2% maximum reflectivity at the same wavelength. However, at λ = 6.63 nm the reflectivity of La_2_O_3_/B_4_C is 39.2%. Such strong reflectivity reliance on the wavelength is ascribed to the fast change in boron optical behavior in the neighborhood of the K_α_ absorption edge at 6.57 nm. They observed that the bandwidth of the reflectivity curves was ~20% lower than that obtained in the ideal structures. This demonstrates an active commingling of the layer materials and a broad thickness of the transition layer between lanthanum and boron carbide-based layers. Moreover, the La_2_O_3_/B_4_C multilayer showed a greater level of imperfections, leading to much-reduced performance.

In an attempt to replace B_4_C with a more boron-rich B_9_C, Andreev et al. obtained 38% reflectivity for La/B_9_C PMMs, which is lower compared to the reported 44% reflectivity of the La/B_4_C multilayer, implying that more boron-rich carbide does not translate to higher reflectivity [53]. It also suggests that variations in the growth behavior of the different materials can have a significant impact on optical performance and may reverse any theoretically predicted enhancements [28]. The major reason for the variation between the experimental and the theoretical reflectance of PMMs has been identified as the expansion of the boundaries in the multilayer structures because of the mixing of the materials.

Lanthanum was shown to form compounds with boron and carbon (LaB_6_ and LaC_2_) in La/B_4_C PMMs [54]. In addition, materials such as Mo, Cr, Sn, and C have been experimented with as an anti-diffusion barrier between La and B_4_C [32,53]. To confirm the barrier properties of the mentioned materials, Andreev and co-workers analyzed the interaction between La/Cr/B_4_C, La/Mo/B_4_C, and La/Sn/B_4_C with periods of 7–8nm (N = 60) and approximately the same layer width ratio (b ≈ 0.5). Their results showed that La/Sn failed because Sn is prone to gathering into drops during the production process. In addition, the reflection coefficients at a 6.69 nm wavelength using Mo and Cr as anti-diffusion barriers are 9% and 4.2%, respectively. These results are significantly lower than those obtained using La/B_4_C without the barriers. The authors explained that the dielectric constants of Cr/La and Mo/La layers are different from the tabulated values of the bulk materials, which may be due to the high chemical activity of La.

Chkhalo et al. compared the effect of carbon as an anti-diffusion barrier [32]. They deposited carbon layers (0.25–0.3 nm thickness) on top of the B_4_C films to prevent direct contact between La and B_4_C. The results of La/B_4_C and La/B_4_C/C PMMs Bragg-reflectivity measurements near the 6.7 nm wavelength are given in Figure 8. The La/B_4_C/C with carbon acting as anti-diffusion layers gave the highest reflection coefficient of 58.6% at a 6.661 nm wavelength, whereas the reflection coefficient of normal La/B_4_C is 40% at the same wavelength. Based on the small-angle X-ray diffraction ((Cu k-α) measurements of the La/B_4_C sample (without a carbon anti-diffusion barrier), the obtained interface parameters are as follows; thickness period d = 3.35 nm, and the portion of La in the period γ_La_ is d_La_/d = 0.5. The width of the La-on-B_4_C transitional region is ~0.75 nm, and the B_4_C-on-La region is ~0.35 nm. The densities of the materials are ρ_La_ = 5.40 g cm^–3^ and ρ_B4C_ = 1.8 g cm^–3^ (for the tabulated values ρ_La_ = 6.17 g cm^–3^ and ρ_B4C_ = 2.0 g cm^–3^). The theoretical calculation also proved that the peak reflectivity of a mirror with the same parameters at the 6.661 nm wavelength should be 40%.

#### Nitridation of Lanthanum Based PMMs

Interfaces in PMMs, for instance B_4_C/La optics for B-K_α_ reflectivity (λ = 6.65 nm), have been shown to go through surface isolation and exothermic interlayer formation through 7La+6B_4_C→4LaB_6_+3LaC_2_ [54]. Moreover, vapor/sputter deposition of B_4_C happens in separate B and C atoms [54,55], enhancing the B_4_C-on-La interface reactivity. Therefore, nitridation is employed to achieve chemically inactive and high contrast interfaces, which are essential in PMMs structures.

Looking at the formation enthalpy (ΔH^for^), absorption constant (β), and refractive index (n) at λ = 6.65 nm of the compounds described in Table 2 [56], it is shown that the passivation of the B_4_C/La interface can be achieved by nitridation, which leads to enhanced reflectivity. In addition, nitridation can repress roughening via grain formation [57] and diffusion at high temperatures. The values of ΔH^for^ suggest that the B_4_C/LaN and BN/LaN interfaces are chemically dormant and are not affected by LaB_6_ and LaC_2_ interlayer formation. IMD modeling and analysis of multilayer films [56,58] show that a 20% increase in peak reflectivity and bandwidth can be obtained in comparison to a 200 period B_4_C/La PMMs when 0.3 nm of both the top and lower side of the La layer is replaced by LaN in a B_4_C/LaN/La/LaN PMMs.

Tsarfati et al. [56] verified the IMD calculations on the optical and kinetics properties by studying nitridation and chemical reactivity in La and B_4_C layers. Their analysis of B_4_C and La layer nitridation by N^2+^ remedy proved that the B_4_C/La interface gradient can be largely decreased by nitridation of the interface. From their bilayer observations, they concluded that nitridation can boost layer-by-layer growth, both through chemical passivation and surfactant-mediated growth of diffusing N_2_ that is weakly bonded in dinitrogen complexes. In their explanation, the loosely bonded N or N_2_ in the B_4_C or La substrate layers partly diffuses into the adlayer, causing surfactant mediated adlayer growth. Successive nitridation of the adlayer was seen to yield nitridated interfaces that are chemically inactive to LaB_6_ and LaC_2_ interlayer formation. B_4_C was observed to swell remarkably upon nitridation, whereas the B content was lowered significantly. The optimum performance was obtained when the La and B_4_C, or only the La layers, were post-N-treated. Their experimental results were extrapolated using IMD modeling, which suggests ~51% peak reflectivities at λ = 6.72 nm for a 200 period multilayer.

Makhotkin et al. [28] investigated the influence of two modes of nitridation on the optical performance. Herein, they employed magnetron sputtering because it is the frequently used technique for the deposition of multilayers consisting of a large number of periods. Two approaches—reactive magnetron sputtering of lanthanum in a blend of argon and nitrogen (represented as La(N)/B) and post-treatment of the La layers by N_2_-ions (defined as La/N/B)—were used to nitridate lanthanum layers. La(N)/B and La/N/B PMMS having 175 periods were deposited to examine the obtainable normal incidence reflectance at a ~6.7 nm wavelength. In both cases, they observed that the nitridation of La significantly increased the optical contrast between La and B, and in turn increased the EUV reflectivity. However, the nitridation did not alter the total width of the interfaces, leaving enough room for further enhancement. In addition, the optical contrast of LaN/B multilayer stacks was limited by the presence of N atoms in the B layers. The normal incidence reflectivities of La(N)/B and La/N/B PMMs are 53% and 57.3%, respectively. In both cases, the measured reflectivity is less than the 60% value shown by their simulations.

Moreover, they observed that the reflectance plot of the La/N/B PMMs revealed a significantly bigger width than the simulated plot and the measured La(N)/B multilayer. Furthermore, La(N)/B PMMs with 57.3% reflectivity is closer to the predicted value (60%), implying that the La(N)/B multilayer deposition possesses lower aperiodicity and no severe growth of the interface roughness. The higher aperiodicity of La/N/B PMMs can be interpreted by the disparity in the deposition process. The post-N-ion treatment is an extra step in the deposition operation. This may result in additional instability of the growth process, which is most visible for a multilayer coating with a high number of periods.

Generally, complete passivation of LaN is preferred in order to maximally prevent admixing between layers. This requires that all the free chemical bonds of lanthanum are filled with nitrogen so that the chances for interaction with the neighboring boron layers are decreased [10]. It is acknowledged that the passivation of lanthanum via the magnetron deposition process in the N atmosphere can vigorously preserve the B-on-La interface. However, there is a risk of the formation of a BN compound at the La-on-B interface, and the chance of obtaining LaB_x_ at that interface cannot be eliminated. As demonstrated by Kuznetsov and co-workers in Figure 9a [10], at the early stage of the La layer deposition in a nitrogen environment, there is a high possibility that both La and N atoms first contact with the boron atoms of the substrate layer rather than the formation of LaN. From Table 1, it is shown in the enthalpies of formation values that the formation of LaB_6_ and BN is thermodynamically feasible.

From the above observations, Kuznetsov designed a structure that eliminates the direct contact of N species with the underlying B layer at the LaN-on-B interface. This was achieved by simply introducing an initial delay in lanthanum nitridation. This resulted in a lanthanum interlayer at the LaN-on-B interface. Herein, the chemical interaction of La with the underlying B layer forms a LaB_x_ interlayer. To confirm the response, they modeled the reflectivity of multilayers structures with varying thicknesses of LaB_6_ interlayers compared to the influence of BN interlayers at the LaN-on-B interface considering effective roughness/diffusion zones of 0.5 nm. The results proved that BN interlayers resulted in a very high reflectivity decrease compared to LaB_6_ (Figure 9b). Furthermore, the synthesized B\La\LaN (the materials are written in the deposition array) PMMs gave a reflectivity of 64.1% at λ = 6.65 nm taken at 1.5° off-normal AOI, as seen in Figure 9c. The increase in reflectance can be ascribed to forming a more optically favorable LaB_6_ instead of BN at the LaN-on-B interface.

## 4. Thermal Stability of the BEUV PMMs

Thermal stability of lithographic multilayer optics for a ~7 nm wavelength is an essential prerequisite for performance because the optics are normally exposed to greater thermal loads (or higher power densities) than the optics used in EUV lithography sources (λ = 13.5-nm) [59]. Exposure to high flux or protracted photons contributes to thermal loading, which may lead to atomic interdiffusion and successive formation of compounds at the interfaces of multilayers [32,60]. This can cause the deterioration of the optical contrast of the multilayer, resulting in decreases in the reflectance [59]. Furthermore, the period-thickness of the multilayer may change due to the formation of compounds, thereby causing an imbalance between the target wavelength and the multilayer period [29]. Nanoscale multilayers are mainly susceptible to slight structural changes at the interfaces. Khorsand et al. [61] exposed Mo/Si multilayers to excessive femtosecond pulse EUV sources. They observed an ultrafast formation of molybdenum silicide because of the intensified atomic diffusion in melted Si, leading to irreversible structural adjustment. The damage mechanism is akin to that seen during thermal treatment as demonstrated by Nedelcu and co-workers [62]. Their study showed growth of silicide interfaces as Mo/Si multilayers are annealed up to 300 °C.

To better understand the effect of thermal stability on PMMs, Naujok and co-workers [63] investigated the change in the microstructure and reflective properties of La/B_4_C and LaN/B_4_C PMMs prepared for high reflectivity at λ = 6.7 nm and annealed at elevated temperatures ≤800 °C. They analyzed the thermally induced changes in the internal structure of the PMMs, such as the elemental distribution, period thickness, crystallinity, and optical reflectivity. At temperatures ≤300 °C, there was no noticeable change in the period thickness of La/B4C PMMs. However, at temperatures >300 °C, there was a significant decrease in the period thickness. This decrease in period thickness was ascribed to the LaB_6_ crystallites’ formation and growth. Further investigation proved that the La/B_4_C multilayer changes to a LaB_6_/C multilayer after annealing at 800 °C for 10 h. This resulted in a shift in wavelength and substantial deterioration of EUV peak reflectance. In Figure 10, it is shown that the initial reflectivity of 49.8% at λ = 6.7 nm, and 8° AOI after deposition decreased to 37.2% at λ = 6.68 nm after annealing at a 400 °C temperature, and further decreased to 2.3% after 800 °C thermal treatment. The decrease in the optical performance was ascribed to the presence of rougher interfaces as a result of the formation of LaB_6_ crystallites. In confirmation, they simulated a LaB_6_/C multilayer considering an average interface width of 0.6 nm, and the reflectivity was shown to be ~3%. For the LaN/B_4_C PMMs, they reported a linear increase in the period thickness at up to a 600 °C temperature. The period thickness further expanded as the temperature increased. They concluded that the increase in the period thickness results from the formation of amorphous BN at the interfaces of the multilayer. Additional studies confirmed no sign of crystallinity throughout the thermal treatment temperature range, and large amorphous LaB_x_C_y_ layers were divided by thin amorphous BN layers. Moreover, their observation of LaN/B_4_C PMMs showed a reduction in reflectivity and significant spectral shift in the peak wavelength, as shown in Figure 10b. The EUV reflectance decreased from the initial value of 57% (λ = 6.7 nm) to 50.2% after a 400 °C annealing temperature. This signifies a 12% reflection loss compared to the 25% loss seen in La/B_4_C PMMs. In addition, when the annealing temperature was 800 °C for 10 h, the EUV reflectance was shown to be 12.6% (λ = 6.96), corresponding to a 20% loss in reflectance. Interestingly, the researchers previously demonstrated that LaN/B PMMs revealed a 20% reflection loss at a 400 °C annealing temperature. Therefore, LaN/B_4_C experienced minimal thermally induced reflection loss compared to the La/B_4_C and LaN/B PMMs.

Nyabero et al. [29] used pulsed DC magnetron sputtering to deposit 50 period Mo/B_4_C multilayers onto neatly polished Si substrates. The thickness of Mo layers was 3 nm, and the thickness of the B4C layers was varied from 1 to 4 nm. The samples were annealed up to 300 °C for 48 h to enable significant intermixing, and the physical causes were analyzed. Their studies showed both expansion and compaction of the period thickness of the multilayers. Figure 11a revealed that the PMMs with B_4_C < 2 nm exhibited compaction, whereas the PMMs with B_4_C > 2.5 nm showed expansion. Moreover, PMMs with B_4_C = 2 nm and 2.5 nm expanded initially before showing compaction. These results proved that the change in the period thickness of Mo/B_4_C PMMs depends significantly on the thickness of the B_4_C layers and the annealing time. Although they observed intense stress relaxation at the thermal treatment time, as seen in Figure 11b, it was shown that the contribution of stress relaxation to the expansion of the period thickness is negligible. Moreover, on the effect of the interdiffusion of atoms during annealing, in PMMs with B_4_C ≤ 1.5 nm, the additional supply of Mo into the earlier formed MoB_x_C_y_ (Figure 11c) interlayer was prevalent, leading to densification, and hence resulting in period thickness compaction. However, for PMMs with B_4_C ≥ 2 nm, it was reported that the higher diffusion of B and C (the mobilities of B and C atoms are higher than Mo atoms) into the interlayers led to the formation of low-density compounds, resulting in a thickness period expansion.

In place of Mo, Zhu et al. [34] used Mo_x_C_1-x_ to grow 100 bilayer Mo_x_C_1-x_/B_4_C PMMs on Si substrate to compare the effect of stress and thermal treatment. After repeated thermal treatment from 100 to 600 °C, the Mo/B_4_C multilayers showed a ∼2% decrease in reflectivity near-normal incident, whereas Mo_x_C_1-x_/B_4_C multilayers proved to be more stable with a 1.4% decrease in reflectivity. However, the ~11% reflectivity value of the as-prepared Mo_x_C_1-x_/B_4_C multilayers made it less competitive in the choice of optical performance. In addition, Rao et al. [64] demonstrated W/B_4_C PMMs with high thermal stability after 800 °C annealing temperatures. They observed that there was no formation of tungsten carbide or tungsten boride during the thermal treatments. In addition, they obtained a near-normal incidence soft x-ray reflectivity of ∼8.3% at λ = 6.8 nm, which decreased to ~7% after annealing at 800 °C, indicating no architectural changes within the layers throughout the thermal treatments. Although the W/B_4_C multilayer offered a thermal stability advantage, its theoretical and experimental reflectivity is very low compared to the other BEUV PMMs we have reviewed so far.

## 5. Summary

EUV lithography has been the dominant candidate for the manufacturing of the next-generation integrated circuits. However, the challenging demands of the semiconductor industry have made the BEUV lithography at ~7 nm wavelength the outstanding candidate for patterning smaller complex features with photons [65,66]. The choice of the ~7 nm wavelength was based on the following factors: ~7 nm is in the range of anomalous dispersion of the optical constants of boron (K-absorption edge at 6.63 nm wavelength). It is expected that boron-containing mirrors should yield high reflection coefficients. Theoretically, La/B PMMs yield a reflectivity of >80% at normal incidence. Moreover, La/B_4_C was first found to possess a theoretical reflectivity of 70% and experimental reflectivity of >30% at normal incidences.

In this review, different factors were considered as major contributors to the variations between the theoretical and experimental reflectivities of the different PMM models. Among these is the enlargement of the interface boundaries of the multilayer coatings due to the intermixing between layers, resulting in a lower reflection peak of the PMMs. One of the remedies is the use of anti-diffusion barriers to suppress the expansion of the interface, and carbon has been identified as the leading material for the anti-diffusion barriers. In addition, nitridation of lanthanum can produce chemically inactive and high contrast interfaces, which will significantly reduce the intermixing between La and B layers in La/B and La/B_4_C PMMs. Furthermore, high interface roughness can cause reduced reflectivity. The interface roughness can be decreased by controlling the deposition temperature, pressure, and thickness of the individual layers. In addition, ion polishing has been suggested to reduce the interface roughness. This involves an in situ etching of the films amid growth using low-energy ions. Ion etching can decrease the surface roughness and increase the optical contrast between interface boundaries through the removal of a top layer with a lesser density that would allow interdiffusion [32]. After identifying the structural limitations of PMMs and proffering solutions to optimize the experimental peak reflectivity of PMMs at a ~7 nm wavelength, the gap between the measured and calculated values remains wide, particularly for Mo/B and La/B PMM models. Therefore, more research and concerted efforts are needed to efficiently design multilayer mirrors with very high reflection at a ~7 nm wavelength.

## Figures and Tables

**Figure 1 nanomaterials-11-02782-f001:**
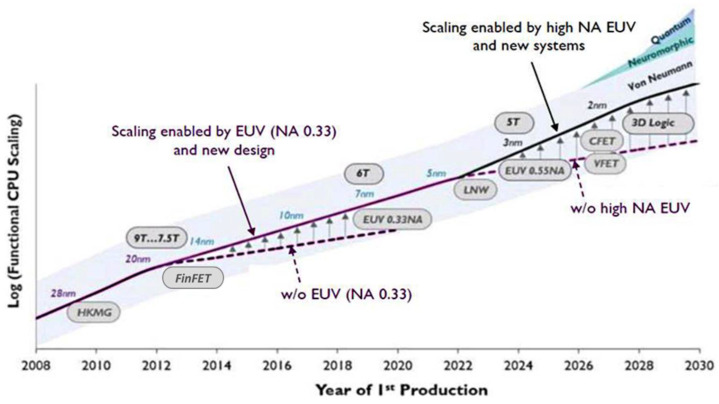
Technology node scaling is driven by the development of lithography. EUV will lead the industry in the next decade. Reprinted from [7], courtesy of IMEC.

**Figure 2 nanomaterials-11-02782-f002:**
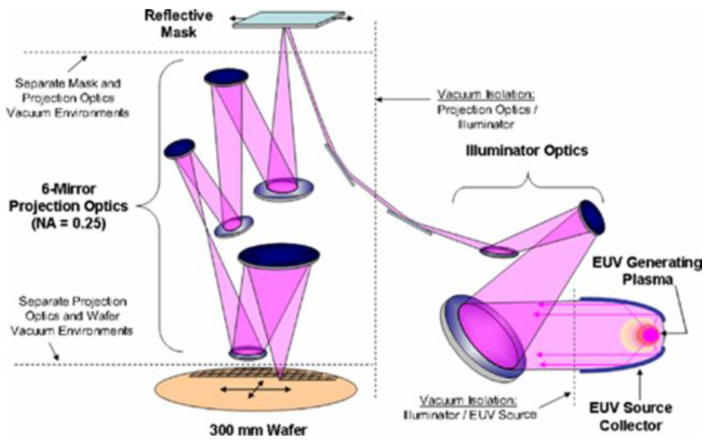
Schematics of the optical system of a lithographical stepper machine. Reprinted from [15].

**Figure 3 nanomaterials-11-02782-f003:**
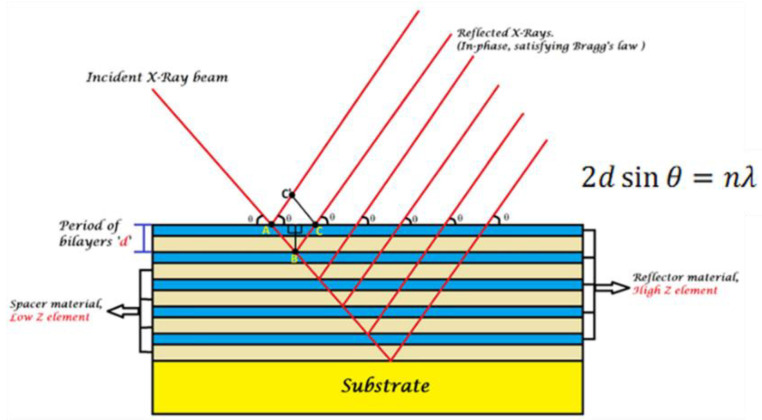
Schematic view of constructive interference from interfaces of a multilayer.

**Figure 4 nanomaterials-11-02782-f004:**
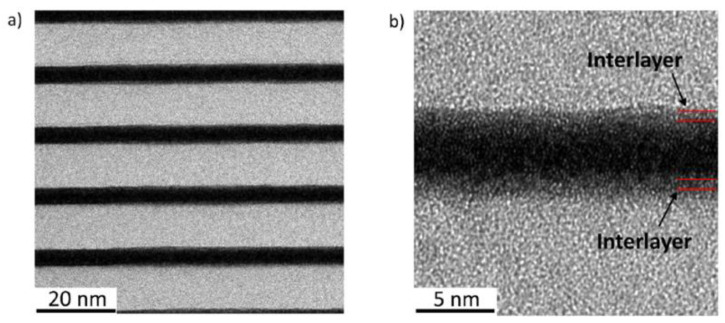
Illustration of intermixing in PMMs. (**a**) Cross-sectional high-resolution TEM image of Mo/B PMMs; (**b**) magnification of the central area. Reprinted from [44] with permission from Elsevier.

**Figure 5 nanomaterials-11-02782-f005:**
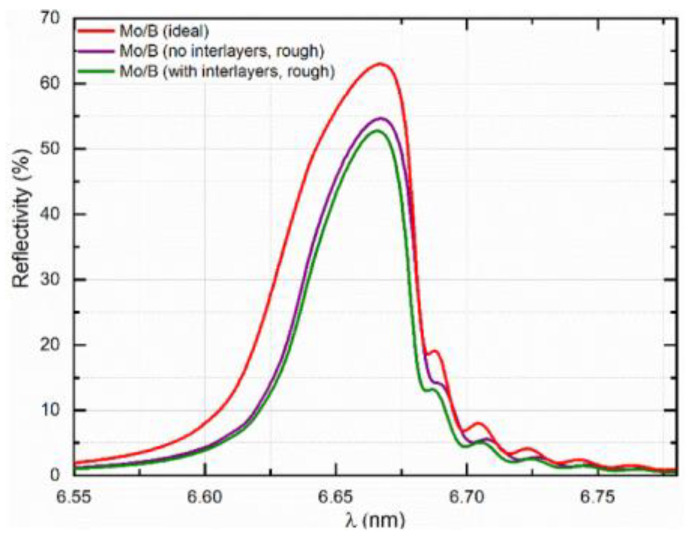
An analogy of the simulated peak reflectivity of varying PMM models at λ = 6.7 nm and 5° off normal angle of incidence: ideal Mo/B multilayer; Mo/B multilayer with interface roughness; real Mo/B multilayer with interlayers. Reprinted from [44] with permission from Elsevier.

**Figure 6 nanomaterials-11-02782-f006:**
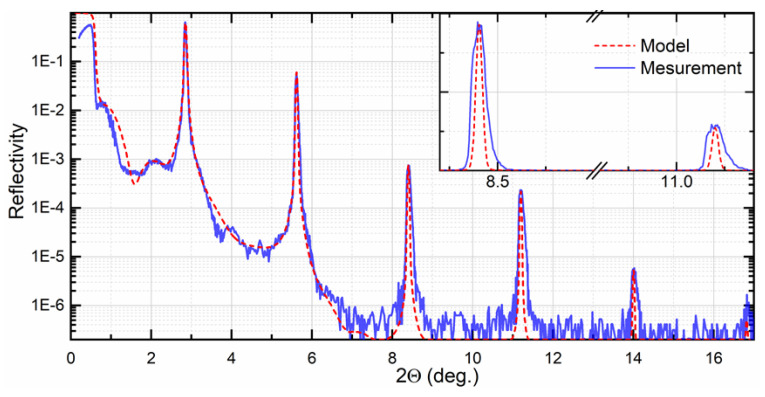
Comparison of measured and calculated low-angle reflectivity curves for Mo/B PMMs having 250 pairs. The inset shows the broadening of the diffraction peak. Reproduced from [44] with permission from Elsevier.

**Figure 7 nanomaterials-11-02782-f007:**
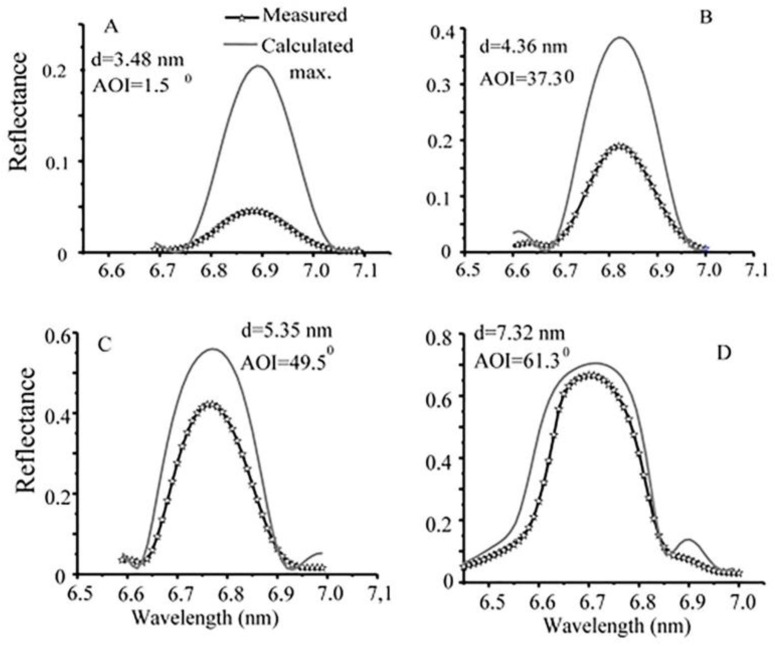
Calculated and measured BEUV reflectivity for e-beam deposited 40 period La/B PMMs with different periods. Reprinted with permission from [28] © The Optical Society.

**Figure 8 nanomaterials-11-02782-f008:**
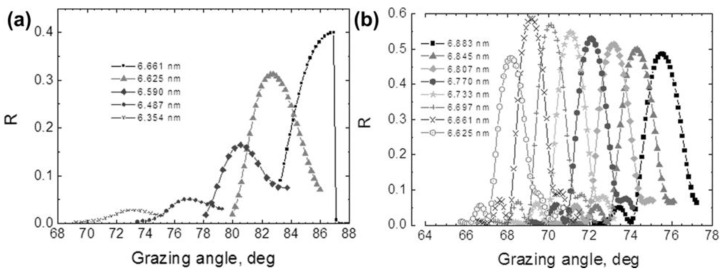
Angular dependence of reflectivity of (**a**) the La/B_4_C, and (**b**) La/B_4_C/C PMMs taken in the spectral range of 6.6–6.9 nm wavelengths. Reprinted from [32] with the permission of AIP Publishing.

**Figure 9 nanomaterials-11-02782-f009:**
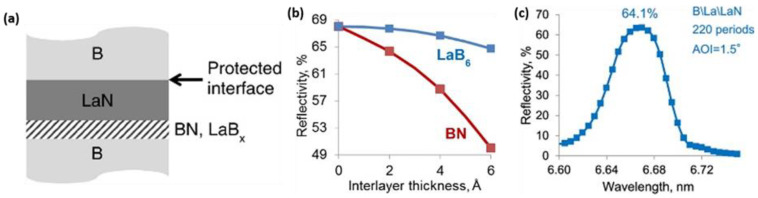
(**a**) Simplified drawing of a La/B multilayer with nitridated lanthanum. The B-on-LaN interface is secured from chemical interaction, whereas BN and LaB_x_ can develop at the LaN-on-B interface. (**b**) Modeled peak reflectivity of the LaN/B multilayers with BN and LaB_6_ as interlayers on the LaN-on-B interface. λ = 6.65 nm, AOI at 1.5° off-normal incidence. (**c**) Measured optical reflectivity of a 220 period B\La\LaN PMM with La interlayers of 0.3 nm thickness introduced at the LaN-on-B interface. The measurement was undertaken at 1.5° off-normal AOI. Reprinted with permission from [10] © The Optical Society.

**Figure 10 nanomaterials-11-02782-f010:**
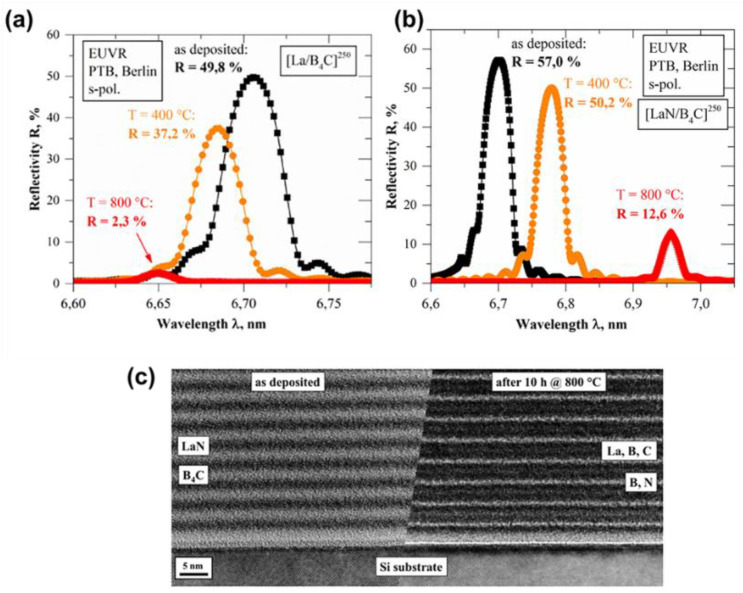
(**a**) EUV spectral of the La/B_4_C PMMs after deposition and after annealing at 400 and 800 °C. (**b**) EUV spectral of the LaN/B_4_C PMMs after deposition and after annealing at 400 and 800 °C for 10 h. (**c**) High-resolution TEM cross-sectional images of the LaN/B_4_C PMMs after deposition (left) and after annealing at 800 °C (right). Reprinted from [64] with permission from Elsevier.

**Figure 11 nanomaterials-11-02782-f011:**
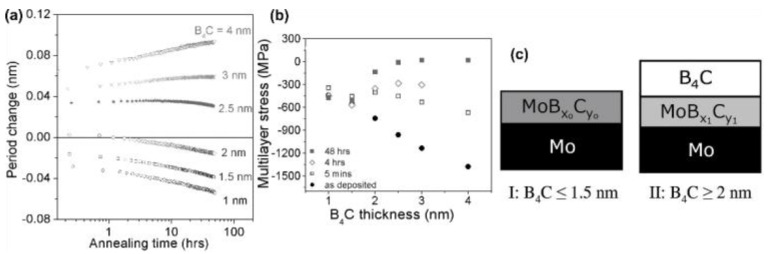
(**a**) Change in the thickness period to the annealing time at 300 °C in the Mo/B_4_C PMMs. (**b**) The stress in the Mo/B_4_C PMMs to the B_4_C thickness at the as-deposited state and different annealing times. (**c**) Schematic depiction of the structural differences in as-deposited Mo/B_4_C PMMs with thicknesses. It is shown that, before thermal treatment, the PMMs with B_4_C ≤ 1.5 nm would possess no or very little pure B_4_C, whereas the PMMs with B_4_C ≥ 2 nm would still possess pure B_4_C. Reprinted from [29] with the permission of AIP Publishing.

**Table 1 nanomaterials-11-02782-t001:** Reported reflectivity of various PMMs for BEUV range.

PMMs	Theoretical Reflectivity	Real Reflectivity	Drawbacks	Reference
LaN/B	75%	64%	Low time stability, manufacturing complexity	Kuznetsov,2015 [10]
Mo/B4C	48%	10%	Low reflectivity	Zhu, 2020 [34]
Mo/B	65%	53%	No experimental confirmation	Penkov, 2021 [44]
La/B4C	69.7%	54.4%	Low time stability	Naujok, 2015 [27]

**Table 2 nanomaterials-11-02782-t002:** The values of ΔH^for^, n, and β (at λ = 6.65 nm) for LaN, BN, LaB_6_, LaC_2_, B_4_C, and La. Reprinted from [56] with permission from Elsevier.

Compound	LaN	BN	LaB_6_	LaC_2_	B_4_C	La
ΔH^for^(KJ/mol)	−303	−255	−130	−89	−71	0
n	0.981	0.995	0.992	0.986	0.999	0.984
β(×10^3^)	1.420	0.894	0.853	0.996	0.528	1.075

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
