# Peer review of "Multilayer Reflective Coatings for BEUV Lithography: A Review"

_nanomaterials, 2021, doi:10.3390/nano11112782_

Round 1
Reviewer 1 Report
The manuscript reviews periodical multilayer reflective coatings (in the title), or periodic multilayer X-ray mirrors (PMMs, as referred in the body text) for BEUV application. The reviews focuses on Mo/B and La(N)/B or B4C multilayers, as best option to work at about 7 nm, which is the desired wavelength with maximum predicted reflectivity. [I think you should decide how to mention such multilayers - periodical reflective coatings or PMMs].
The manuscript is well organized and discusses the findings in the different reported articles pointing at the advantages vs limitations of the different configurations studied so far. The findings are analyzed in terms of theoretical vs experimental results. Attention is paid to point out the possible sources of inconsistencies in terms of thickness control over the period repetitions in the superlattices and on interface width. The latter is discussed in terms of chemical element interdiffusion and topological roughness, both phenomena being relevant in the decrease of the reflectivity.
Overall the manuscript is well organized and written, with minimal typos I could spot.
In my opinion the manuscript can be accepted after minor revision on the following specific points:
line 330 - please introduce IMD in long form the first time it appears
line 381-382 Text reports: "From Table 1, it 381 is shown in the enthalpies of formation values that the formation of LaB6 and BN are thermodynamically favorable."- However, based on Table 1 - LaN has more negative enthalpy of formation than LaB6 and BN suggesting LaN is favored to form. Could you please rephrase or comment to make clear the most favorable compound to form and the competition among LaN, BN and LaB6 to form in La-on-B and B-on-La interfaces when nitridation is used?
Author Response
Response to the review comments
Thank you so much for your insightful contributions, and below are my responses to your comments
Reviewer one
- The manuscript reviews periodical multilayer reflective coatings (in the title), or periodic multilayer X-ray mirrors (PMMs, as referred in the body text) for BEUV application. The reviews focus on Mo/B and La(N)/B or B4C multilayers, as best option to work at about 7 nm, which is the desired wavelength with maximum predicted reflectivity. [I think you should decide how to mention such multilayers - periodical reflective coatings or PMMs].
We adopted PMMs throughout the manuscript, “reflective” is added to the title for emphasis.
- line 330 - please introduce IMD in long form the first time it appears.
IMD is the generic name of the computer program. We have changed “IMD-modeling” to IMD-modeling and analysis of multilayer films and added the reference number 59 (D.L. Windt, IMD—Software for modeling the optical properties of multilayer films, Computers in Physics 12(4) (1998)) to avoid confusion.
3.line 381-382 Text reports: "From Table 1, it 381 is shown in the enthalpies of formation values that the formation of LaB6 and BN are thermodynamically favorable."- However, based on Table 1 - LaN has more negative enthalpy of formation than LaB6 and BN suggesting LaN is favored to form. Could you please rephrase or comment to make clear the most favorable compound to form and the competition among LaN, BN and LaB6 to form in La-on-B and B-on-La interfaces when nitridation is used?
The word “favorable” has been changed to “feasible” for better understanding.
Reviewer 2 Report
The paper is an excellent review about multilayer reflective coatings for BEUC lithography. It is very well ellaborated and in my opinion it deserves to be published.
Author Response
Thank you for your review.